# Boosting Search Engines with Interactive Agents

## Abstract

This paper presents first successful steps in designing agents that learn meta-strategies for iterative query refinement. Our approach uses machine reading to guide the selection of refinement terms from aggregated search results. Agents are then empowered with simple but effective search operators to exert fine-grained and transparent control over queries and search results. We develop a novel way of generating synthetic search sessions, which leverages the power of transformer-based language models through (self-)supervised learning. We also present a reinforcement learning agent with dynamically constrained actions that learns interactive search strategies from scratch. We obtain retrieval and answer quality performance comparable to recent neural methods using a traditional term-based BM25 ranking function. We provide an in-depth analysis of the search policies.

## 1 Introduction

Can machines learn to use a search engine as an interactive tool for finding information? Web search is the portal to a vast ecosystem of general and specialized knowledge, designed to support humans in their effort to seek relevant information and make well-informed decisions. Utilizing search as a tool is intuitive, and most users quickly learn interactive search strategies characterized by sequential reasoning, exploration, and synthesis (Hearst, 2009; Rutter et al., 2015; Russell, 2019). The success of web search relies on machines learning human notions of relevance, but also on the users' ability to (re-)formulate appropriate queries, grounded in a tacit understanding of strengths and limitations of search engines. Given recent breakthroughs in language models (LM) (Vaswani et al., 2017; Devlin et al., 2019; Brown et al., 2020) as well as in reinforcement learning (RL) (Mnih et al., 2013; Silver et al., 2016; Berner et al., 2019), it seems timely to ask *whether*, and *how*, agents can be trained to interactively use search engines. However, the lack of expert search sessions puts supervised learning out of reach, and RL is often ineffective in complex natural language understanding (NLU) tasks. The feasibility of autonomous search agents hence remains an open question, which inspires our research.

We pursue a design philosophy in which search agents operate in structured action spaces defined as generative grammars, resulting in compositional, productive, and semantically transparent policies. Further domain knowledge is included through the use of well-known models and algorithms from NLU and information retrieval (IR). Most notably, we develop a self-supervised learning scheme for generating high-quality search session data, by exploiting insights from relevance feedback (Rocchio, 1971), used to train a supervised LM search agent based on T5 (Raffel et al., 2020). We also build an RL search agent based on MuZero (Schrittwieser et al., 2020) and BERT (Devlin et al., 2019), which performs planning via rule-constrained Monte Carlo tree search and a learned dynamics model.

We run experiments on an open-domain question answering task, OpenQA (Lee et al., 2019). Search agents learn diverse policies leading to deep, effective explorations of the search results. The MuZero agent outperforms a BM25 (Robertson & Zaragoza, 2009) search function running over a Wikipedia index, on both retrieval and answer quality metrics. Thus, providing novel evidence for the potential of knowledge-infused RL in hard NLU tasks. The T5 agent can more easily leverage large pre-trained encoder-decoders and proves superior to MuZero. Furthermore, a straightforward ensemble of agents is comparable in performance to a state of the art neural retrieval system, DPR (Karpukhin et al., 2020), while relying solely on interpretable, symbolic retrieval operations. This suggests new challenges for future work; e.g., involving hybrid architectures and policy synthesis.[1]

---

[1]We open-source the code and trained checkpoints for both agents: anonymized during review.

$q_0$: who won the us open

Figure 1: Schematic agent interaction with the search environment (Lucene-BM25) for an ambiguous query (tennis/golf). After receiving a set of documents ($\mathcal{D}_t$), the corresponding observation ($o_t$) is compiled by ranking all documents $\in \cup_{i=0}^{t}\mathcal{D}_i$ by their Passage Score (PS), and creating snippets for the top-$k$ documents around the answers extracted by the Machine Reader (MR). Note that PS/MR always condition on $q_0$. In step 1, $q_1$, the agent doubles the weight on term 'tennis' in document titles (we use the Lucene query syntax). In the last step, $q_T$, the agent excludes from search results all documents containing the term 'pga' (golf), then outputs the top-$k$ documents found in the session.

## 2 SELF-SUPERVISED LEARNING FOR INTERACTIVE SEARCH

It has been a powerful vision for more than 20 years to design search engines that are intuitive and simple to use. Despite their remarkable success, search engines are not perfect and may not yield the most relevant result(s) in one shot. This is particularly true for rare and intrinsically difficult queries, which may require interactive exploration by the user to be answered correctly and exhaustively. Contextual query refinement is a common technique (Jansen et al., 2009), even among children (Rutter et al., 2015), used to improve search by combining evidence from previous results and background knowledge (Huang & Efthimiadis, 2009). Such refinements often rely on inspecting result snippets and titles or on skimming the content of top-ranked documents. This process is iterative and may be repeated to produce a sequence of queries $q_0, q_1, \ldots, q_T$ until (optimistically) a satisfactory answer is found. It seems natural to mimic this interactive process by a search agent, which learns the basic step of generating a follow-up query from previous queries and their search results.

Furthermore, it is noteworthy how power users apply dedicated search operators and sophisticated investigative strategies to solve deep search puzzles (Russell, 2019). In particular, unary operators offer a great deal of fine-grained control and transparency and as such are highly effective in expert hands. We concentrate on three operators: '+', which limits results to documents that contain a specific term, '-' which excludes results that contain the term, and '$\wedge_i$' which boosts a term weight in the BM25 score computation by a factor $i \in \mathbb{R}$. For instance – see also Figure 1 – the query 'who won the us open', may lead to both tennis and golf-related results. An expert searcher could zero-in on the tennis intent by excluding golf-related terms and boosting tennis-related ones. As we show in this paper, these operators are also pivotal in designing interactive search agents.

### 2.1 RESULT AGGREGATION FOR CONTEXTUAL REFINEMENT

Web searchers expect the best answer to be among the top two hits on the first results page (Hearst, 2009, §5) and pay marginal attention to the bottom half of the *10 blue links* (Granka et al., 2004; Joachims et al., 2005; Nielsen & Pernice, 2009; Strzelecki, 2020). Likewise, a search agent considers only the top $k$ documents returned by the search engine at every step, where $k = 5$. During a search session the agent maintains a list of the top-$k$ documents overall, which is returned at the end.

To aggregate results we use a machine reader (MR, cf. (Rajpurkar et al., 2016)). Specifically, we use a DPR-like reader/passage scorer (Karpukhin et al., 2020), which builds upon a pre-trained BERT model. Beyond identifying the most promising answer span within each result document $d$, the system also estimates the probability of $d$ containing the (unspecified) answer $P(d \ni \text{ANSWER} \mid q) \in [0; 1]$. This probability can be viewed as a Passage Score (PS) that induces a calibrated ranking across all result documents within a session.

An observation representing the session at any given step is built by extracting a fixed-length token window centered at the answer span predicted by the reader for each document. In addition, we include the document titles. Finally, the query tokens and refinements describing $q_t$ are also included. This leads to a segmented observation token sequence $o_t$ which is truncated to length $\leq 512$ , a common input length for pre-trained transformer-based LMs (cf. Appendix B for details, examples). We then use BERT or T5 to produce an embedding $\mathbf{s}_t$ from which the search agent will generate the next query. If we denote the result set for $q_t$ by $\mathcal{D}_t$, then we get diagrammatically

$$
\begin{bmatrix} q_0, \ldots, q_t \\ \text{search} \downarrow \text{engine} \\ \mathcal{D}_0, \ldots, \mathcal{D}_t \end{bmatrix} \underbrace{\overset{\text{MR/PS}}{\longmapsto} o_t}_{\text{observation}} \underbrace{\overset{\text{LM}}{\longmapsto} \mathbf{s}_t}_{\text{encoding}} \underbrace{\overset{\text{agent}}{\longmapsto} q_{t+1}}_{\text{generation}} \tag{1}
$$

We focus on the case where $q_{t+1}$ is obtained from $q_t$ through augmentation. This may add a keyword $w \in \Sigma^{\text{idx}}$, where $\Sigma^{\text{idx}}$ is the search index vocabulary, with the usual disjunctive search engine semantics or a structured search term formed by the use of unary operators ('+','-','$\wedge$') and fields.

## 2.2 ROCCHIO QUERY EXPANSIONS

In the absence of training sessions from human expert users, we propose to generate synthetic search sessions in a self-supervised manner, making use of a set of question-answer pairs $(q, a)$. We initialize $q_0 = q$ and aim to find a sequence of refinements that make progress towards identifying documents containing the answer $a$, based on a reward function $q_t \mapsto \mathcal{D}_t \mapsto r_t \in [0; 1]$ (cf. §4). A query is *not* further refined, if either $t=20$ (maximal length) or if no score increasing refinement can be found.

To create candidate refinements, we make use of the idea of relevance feedback as suggested in Rocchio (1971). An elementary refinement – called a *Rocchio expansion* – then takes the form

$$
q_{t+1} := q_t \, \Delta q_t, \Delta q_t := [+ \mid - \mid \wedge_i \quad \text{TITLE} \mid \text{CONTENT}] \; w_t, w_t \in \Sigma_t := \Sigma_t^q \cup \Sigma_t^\tau \cup \Sigma_t^\alpha \cup \Sigma_t^\beta \tag{2}
$$

where $i$ is the boosting coefficient and $\Sigma_t$ refers to a set of terms accessible to the agent. By that we mean terms that occur in the top PS-ranked session documents. We use superscripts to refer to the vocabulary of the question ($q$), titles ($\tau$), answers ($\alpha$) or bodies ($\beta$) of documents in $o_t$. Note that adding terms $\notin \Sigma_t$ would make refinements difficult to reproduce for an agent and thus would provide supervision of low utility.

Another aspect of creating sessions as described above has to do with the search complexity of finding optimal sequences of Rocchio expansions. We consider $q_* = q + a$ as the "ideal" query, whose results define the vocabulary $\Sigma_*$. For efficiency reasons, we further constrain the terms to be added via exact matches, term boosting or term exclusions by defining respective constrained dictionaries

$$
\Sigma_t^\uparrow = \Sigma_t \cap \Sigma_*, \quad \Sigma_t^\downarrow = \Sigma_t - \Sigma_* . \tag{3}
$$

This means it is possible to upgrade accessible terms, $w_t$, to exact matches, or weight boosting, if they also occur in the ideal result set ($w_t \in \Sigma_t^\uparrow$); and to exclude accessible terms if they are not present in the ideal results ($w_t \in \Sigma_t^\downarrow$). We have found experimentally that this leads to a good trade-off between the quality of Rocchio expansions and the search effort to find them. The search for sequences of Rocchio expansions is done heuristically. More details, pseudo-code illustrating the procedure and examples can be found in §5, Appendix A Appendix G.

## 2.3 SELF-SUPERVISED T5 AGENT

We suggest to train a generative search agent in a supervised manner by making use of synthetic search sessions generated by Rocchio expansions. We use T5, a pretrained transformer encoder-decoder model which achieves state-of-the-art results on multiple NLU tasks. As a search agent, T5 predicts a single new search expansion from an observed state. In the spirit of *everything-is-string-prediction*, both state and expansions are represented as plain strings. See Appendix B for a full example.

Our T5 agent is trained via Behavioral Cloning (BC) (Michie, 1990). We treat each step in a Rocchio session as a single training example. As is common in sequence prediction tasks, we use the cross-entropy loss for optimization. BC is perhaps the simplest form of Imitation Learning (IL), and has been proven effective in a variety of application domains (Sharma et al., 2018; Rodríguez-Hernandez

et al., 2019). In our query refinement task, it allows to inherit the expressive power of the Rocchio query expansions and, differently from other IL approaches (Ross et al., 2011; Ho & Ermon, 2016; Ding, 2020), requires only *offline* interactions with the search engine. Crucially, this enables scaling to the large action spaces and model sizes typical of recent LMs. Our T5 agent can also be described as a Decision Transformer with fixed max return (Chen et al., 2021).

At test time, we start with the initial query and incrementally add new expansions, querying the trained T5 model in every step. We then use the refined query to retrieve new documents and continue until either the set of new documents is empty or we reach the maximum number of steps. Throughout the session, we maintain the top-5 documents among all those retrieved.

## 3 REINFORCEMENT LEARNING: MUZERO AGENT

MuZero (Schrittwieser et al., 2020) is a state-of-the-art agent characterized by a learnable model of the environment dynamics. This allows the use of Monte Carlo tree search (MCTS) to predict the next action, in the absence of an explicit simulator. In our use case, MuZero aims to anticipate the latent state implied by each action with regard to the results obtained by the search engine. For instance, in the example of Figure 1, it may learn to predict the effect of boosting the term 'tennis'. This approach to planning is intuitive for search, as searchers learn to anticipate the effect of query refinements while not being able to predict specific results. Furthermore, this offers a performance advantage of many orders of magnitude against executing queries with the *real* search engine.

### 3.1 GRAMMAR-GUIDED SEARCH

To map observations to states, the MuZero agent employs a custom BERT with additional embedding layers to represent the different parts (cf. Appendix B for details). Compared to T5, MuZero has a more challenging starting point: its BERT-based representation function is pre-trained on less data, it has fewer parameters (110M vs. billions) and no cross-attention: predictions are conditioned on a single vector, [CLS]. Moreover, it cannot as easily exploit supervised signals. However, it can more openly explore the space of policies, e.g. independent of the Rocchio expansions. Through many design iterations, we have identified it to be crucial to structure the action space of the MuZero agent and constrain admissible actions and refinement terms dynamically based on context. This provides a domain-informed inductive bias that increases the statistical efficiency of learning a policy via RL.

We take inspiration from generative, specifically context-free, grammars (CFGs) (Chomsky, 1956) and encode the structured action space as a set of production rules, which will be selected in (fixed) top-down, left-to-right order. A query refinement is generated as follows

$$Q \Rightarrow U\,Q \,|\, W\,Q, \quad U \Rightarrow \text{Op Field } W, \quad \text{Op} \Rightarrow +\,|\,-\,|\wedge_i, \quad \text{Field} \Rightarrow \text{TITLE}\,|\,\text{CONTENT}, \quad (4)$$

which allows for adding plain or structured keywords using unary operators. The selection of each refinement term $W$ proceeds in three steps, the first two can be described by the rules

$$W \Rightarrow W_t^q \,|\, W_t^\tau \,|\, W_t^\beta \,|\, W_t^\alpha \,|\, W^{\text{idx}}, \quad W_t^x \Rightarrow w \in \Sigma_\tau^x, \quad x \in \{q, \tau, \beta, \alpha\}, \quad W^{\text{idx}} \Rightarrow w \in \Sigma^{\text{idx}} \quad (5)$$

which means that the agent first decides on the origin of the refinement term, i.e., the query or the different parts of the top-scored result documents, and afterwards selects the term from the corresponding vocabulary. As the term origin correlates strongly with its usefulness as a refinement term, this allows to narrow down the action space effectively. The agent is forced to pick a term from the larger vocabulary (1.6M terms) of the search index $\Sigma^{\text{idx}}$ during MCTS, as there is no observable context to constrain the vocabulary.

The third level in the action hierarchy concerns the selection of the terms. We have found it advantageous to make use of subword units; specifically, BERT's 30k lexical rules involving word pieces, to generate terms sequentially, starting from a term prefix and adding one or more suffixes. Note that this part of the generation is context-sensitive, as we restrict the generation to words present in the vocabulary. We make use of tries to efficiently represent each $\Sigma_\tau^x$ and amortize computation. The grammar-guided MCTS is explained in detail in Appendix F.

## 4    THE OPENQA ENVIRONMENT

We evaluate search agents in the context of open-domain question answering (Open-QA) (Voorhees, 2000; Chen et al., 2017). Given a question $q$, we seek documents $\mathcal{D}$ that contain the answer $a$ using a search engine, the environment. Following common practice, we use Lucene-BM25 with default settings on the English Wikipedia. BM25 has provided the reference probabilistic IR benchmark for decades (Robertson & Zaragoza, 2009), only recently outperformed by neural models (Lee et al., 2019). The Lucene system provides search operators comparable to commercial search engines.

Exploration-based learning is vulnerable to discovering adversarial behaviors. As a safeguard we design a composite reward. The score of a results set $\mathcal{D}$, given $q$, interpolates three components. The first is the Normalized Discounted Cumulative Gain (NDCG) at $k$. See Eq. 6a, where $w_i = \log_2(i+1)^{-1}/\sum_{j=1}^{k}\log_2(j+1)^{-1}$ are normalizing weights, and $\mathrm{rel}(d|q) = 1$, if $a \in d$, 0 otherwise:

$$a)\ \mathrm{NDCG}_k(\mathcal{D}|q) = \sum_{i=1}^{k} w_i\,\mathrm{rel}(d_i|q), \qquad b)\ \mathrm{NDCEM}_k(\mathcal{D}|q) = \sum_{i=1}^{k} w_i\,\mathrm{em}(d_i|q). \tag{6}$$

NDCG is a popular metric in IR as it accounts for rank position, it is comparable across queries, and it is effective at discriminating ranking functions (Wang et al., 2013). NDCG alone can have drawbacks: on "easy" questions a score of 1 can be achieved in short meritless episodes, while on "hard" ones it may be impossible to find a first valid step, since Eq. 6a takes discrete values. Hence, we introduce a second component, $\mathrm{NDCEM}_k$ (Eq. 6b) where $\mathrm{em}(d|q) = 1$ if the answer extracted from $d$ by the reader exactly matches $a$, 0 otherwise. $\mathrm{NDCEM}_k$ helps validate results by promoting high-ranking passages yielding correct answer spans. Finally, to favour high-confidence result sets we add the normalized Passage Score of the top $k$ results, leading to the following scoring function

$$\mathrm{S}_k(\mathcal{D}|q) := (1-\lambda_1-\lambda_2)\cdot\mathrm{NDCG}_k(\mathcal{D}|q)+\lambda_2\cdot\mathrm{NDCEM}_k(\mathcal{D}|q)+\lambda_1\cdot\frac{1}{k}\sum_{i=1}^{k}\mathrm{PS}(d_i|q) \quad \in [0,1] \tag{7}$$

Based on (7), we define the search step reward

$$r_t = \mathrm{S}_5(\mathcal{D}_t|q_0) - \mathrm{S}_5(\mathcal{D}_{t-1}|q_0). \tag{8}$$

We train the MuZero agent directly on the reward. The reward is sparse, as none is issued in between search steps. The T5 agent is trained indirectly on it via the induction of Rocchio sessions (cf. §2.2).

## 5    EXPERIMENTS

We use the OpenQA-NQ data (Lee et al., 2019), derived from Natural Questions (Kwiatkowski et al., 2019), consisting of Google queries paired with answers extracted from Wikipedia. The data includes 79,168 train questions, 8,757 dev questions and 3,610 for test. We use the provided partitions and Wikipedia dump. Following Lee et al. (2019) we pre-process Wikipedia into blocks of 288 tokens, for a total of 13M passages. We evaluate each system on the top-5 288-token passages returned. Model selection and data analysis are performed on NQ Dev, using the reward (Eq. 8) as the objective.

We generate synthetic search sessions using Rocchio expansions for 5 grammars: **G0** (only simple terms), **G1** (only term boosting), **G2** ('+' and '-'), **G3** (G0+G2) and **G4** (G0+G1+G2). We use the reward above with $\lambda_1=0.2, \lambda_2=0.6$, after a search against the quality metrics (cf. Appendix C). We select 5 possible values, $i \in \{0.1, 2, 4, 6, 8\}$, for term boosting weights. At each step, we attempt at most $M$ BM25 searches, based on the current observation $o_t$, to find an expansion that improves the reward. To further improve efficiency, we sort the terms in $o_t$ by Lucene's IDF score and keep the top $N$. We set $N=M=100$. For NQ Train, and G4, we find 298,654 Rocchio expansions from 77,492 questions. Similarly, we generate sessions for the NQ Dev and Test to estimate headroom.

### 5.1    AGENTS TRAINING AND INFERENCE

The DPR reader/selector and MuZero's $h_\theta$ function use 12-layer BERT systems.[2] To train the former, we generate for each query in NQ Train 200 candidate passages from our BM25 system, picking

---

[2]BERT-base, initialized from `https://tfhub.dev/google/bert_uncased_L-12_H-768_A-12/1`.

Table 1: Results on NQ Test. For DPR Top-{1,5} performance we report the most recent numbers, the published performance Karpukhin et al. (2020) is slightly lower (47.3 and 68.1, respectively).

| Metric | BM25 | PS($\mathcal{D}_0$) | MZ | T5-G1 | MZ+T5s | DPR | RQE-G4 |
|--------|------|------|------|------|------|------|------|
| NDCG@5 | 21.51 | 24.82 | 32.23 | 44.27 | 46.22 | - | 65.24 |
| Top-1 | 28.67 | 44.93 | 47.97 | 52.60 | 54.29 | 52.47 | 73.74 |
| Top-5 | 53.76 | 53.76 | 59.97 | 66.59 | 71.05 | 72.24 | 88.17 |
| EM | 28.53 | 41.14 | 32.60 | 44.04 | 44.35 | 41.50 | 62.35 |

one positive and 23 negative passages for each query at random whenever the query is encountered during training. The reader/scorer is not trained further.

The MuZero implementation is scaled and distributed via an agent-learner setup (Espeholt et al., 2018) in the SEED RL (Espeholt et al., 2020) framework allowing for centralized batching of inference for effective use of accelerators. MuZero is trained on NQ Train for a total of 1.6 million steps ($\approx$10 days) using 500 CPU-based actors and 4 Cloud TPU v2 for inference and training on the learner.[3] For each step, 100 simulations are performed. During training, we limit sessions to a maximum of 20 steps. The agent also can decide to stop early by selecting a dedicated stop action. Training of MuZero can be improved by providing *advice* to the actors. An actor may receive information about which terms $w_t$ should be promoted, $w_t \in \Sigma_t^\uparrow$, or demoted, $w_t \in \Sigma_t^\downarrow$. The probability of an episode receiving advice starts at $0.5$ and decays linearly to 0 in one million steps.

For the T5 agent we start from the pretrained T5-11B (11 billion parameters) public checkpoint and continue training on the NQ Train expansions. Training took about 5 days using 16 Cloud TPU v3. At inference time, we found that fixing the sessions to 20 steps worked best for both T5 and MuZero. We report detailed training configurations and ablations in Appendix D.

## 5.2 RESULTS

Table 1 summarizes the results on NQ Test. We evaluate passage retrieval quality by means of ranking (NDCG@5) and precision (Top-1, Top-5) metrics. We also report Exact Match (EM) to evaluate answer quality. Our baseline is Lucene's BM25 one-shot search. Reranking the same BM25 documents by the PS score (PS($\mathcal{D}_0$)) is easy and improves performance on all metrics, particularly noticeable on Top-1 and EM,[4] providing a more informative comparison. The last column (RQE-G4) reports the metrics of the episodes generated by the Rocchio Query Expansion process (§2.2) using the grammar with all operators (G4). RQEs use the gold answer and can be seen as a, possibly conservative, estimate of the performance upper bound. As the external benchmark we use DPR (Karpukhin et al., 2020), a state-of-the-art OpenQA neural retriever based on dual encoders, the dominant architecture for deep learning-based ad hoc retrieval (Craswell et al., 2020).

The MuZero agent (MZ) outperforms both BM25 and PS($\mathcal{D}_0$). While this result may seem trivial, it marked a milestone that required many iterations to achieve. The challenges for RL in IR, and NLU, are extreme in terms of state and action space dimensionality, data sparsity etc. (Zhang et al., 2021). We propose ideas for tackling some of these key challenges by fitting out agents with domain knowledge in principled ways, with the grammar-guided MCTS as the centerpiece, The best MuZero converges to a policy which uses only term boost actions with a weight of 2 (Figure 2a). This agent is not able to find better-performing, diverse policies. This is an extreme case of a more general pattern. Different sub-grammars represent different tactics; e.g., '+' and '-' affect the accessible documents in irreversible ways, while boosting only affects ranking. It is challenging for all agents, and particularly MuZero, to modulate effectively multiple sub-policies.

Training T5 is less involved than MuZero. This allows us to evaluate all 5 grammars. The best one, 'T5-G1' in Table 1, is limited to term boosting (G1), but it learns to use all available weight values (Figure 2a). In terms of Top-1 this agent outperforms the published and the most recently posted DPR results[5] but has worse Top-5 than both. Results for all five T5 agents are found in Table A.5.

---

[3]For details, see https://cloud.google.com/tpu.

[4]Top-5 is identical to BM25 since the documents are the same.

[5]https://github.com/facebookresearch/DPR.

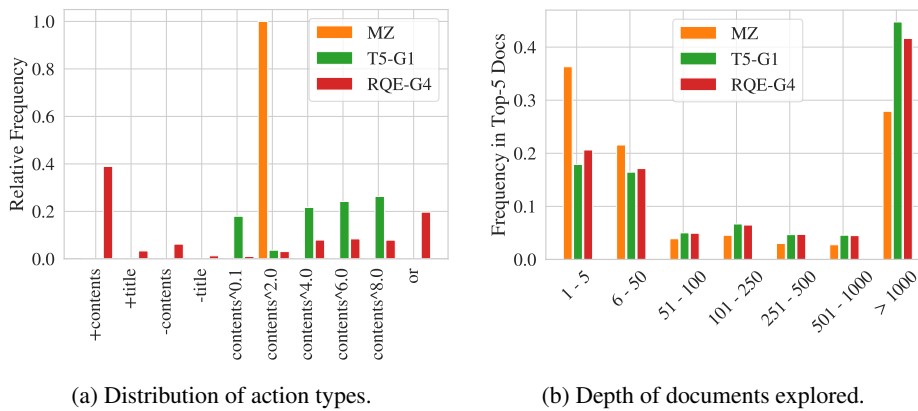

(a) Distribution of action types.          (b) Depth of documents explored.

Figure 2

In the last experiment we combine all trained agents, the five T5 agents and the MuZero one, in one ensemble. We simply rank the union of all the documents returned by the ensemble, by means of the PS score on each document. This ensemble ('MZ+T5s' in Table 1) has slightly better precision than the recent DPR in top position, and slightly worse for the Top-5. This results indicates that the ability to orchestrate diverse sub-policies may indeed be the key to future progress for search agents. The current SOTA for Top-5 is 74.0 (Qu et al., 2021).

We conclude by discussing answer quality. Agents routinely produce answer spans, as predicted by the reader/scorer, to build observations. The MR/PS component is trained once, before the agents, on the output of BM25. However, agents deeply change results. As Figure 2b shows, they dig deep in the original BM25 ranking. This is positive, as behavior discovery is one of the main motivations for researching exploratory methods like RL. As a consequence, though, the MR/PS component effectively operates out of distribution and the EM numbers of the internal reader are not competitive with recent methods, Table A.6 reports all the numbers including on NQ Dev.

Ideally, one would co-train the observation builder with the search agent. However, combining the two would introduce significant engineering complexity in the current architecture. For instance, one could interleave training the two as in DQNs (Mnih et al., 2013). A simpler alternative is to add the answer prediction task to the T5 agent. Retrieval-augmented answer generation is known to produce strong results (Izacard & Grave, 2021). Multitasking would simplify the design of the generative agents and possibly produce better models. We make a first step in this direction by training one single, dedicated T5 agent. The system uses as training input the top-5 documents of the RQE-G4 episodes, but its task is to generate the gold answer, instead of the query expansion. By using the output of the 'T5-G1' and 'MZ+T5s' agents, the EM performance of the answer generation T5 is comparable to methods that build on DPR, such as RAG (Lewis et al., 2020b) (44.5 EM). Although not as good as FID (Izacard & Grave, 2021) that condition on many more (100) documents. The performance of the MuZero agent is lower here, we believe this is due its more idiosyncratic results.

## 5.3 DISCUSSION AND FUTURE WORK

Table 2 illustrates an example where the T5-G4 agent switches policy mid-session. This agent is trained on the full grammar (G4) and can use all search operators. The question is about basketball records and BM25 does not find good results. In the first three steps the agent focuses on *re-ranking* by boosting terms like 'per' (from the phrase 'per game' in the results for $q_0$) and 'scorer'. This produces a good hit and answer span ('Pete Maravich') at position 1 of step 3. The agent then switches to *filtering* mode, to zero in on documents containing the predicted answer term. This is a clear instance of successful policy synthesis. However it is a gamble, as predicted answers can be incorrect and filtering actions are not reversible. This suggests that the action space may benefit by including more control actions, e.g. to undo or go back to a specific state, to better support safe exploration and the emergence of meta policies. We plan to investigate this in future work.

Table 2: Example of a T5-G4 agent session exhibiting multiple tactics. We only report three steps and the top-3 results, for brevity. In red the answer spans wrongly predicted by the internal reader (in blue when correct). The score for the result set at each step is that of Equation 7.

| $\mathbf{q_0}$ | who averaged the most points in college basketball history | [**score:** 0.027] |
|---|---|---|
| $\mathbf{d_1}$ | Title: Gary Hill (basketball) 
 . . . one of four on that team who averaged double figures in points. Senior **Larry Jones** was OCU's leading scorer at 19.7 points a game, sophomore Bud Koper added 15.9. . . | |
| $\mathbf{d_2}$ | Title: Kevin Foster (basketball) 
 . . . his senior year, Foster averaged 21 points per game and was named the MVP and All-District 18-5A First Team. He was also a Texas top- **30** player his final season . . . | |
| $\mathbf{d_3}$ | Title: Paul Cummins (basketball) 
 . . . big home win over Army. As a freshman, Cummins high-scored with **13** points against final-four team Louisville (2004). After graduating in 2008, Cummins played for . . . | |
| $\mathbf{q_3}$ | '$\mathbf{q_0}$' (contents:"per"∧6) (contents:"scorer"∧4) (contents:"3"∧6) | [**score:** 0.330] |
| $\mathbf{d_1}$ | Title: Alphonso Ford 
 . . . seasons. With 3,165 career points scored in the NCAA Division I, he is 4th on the all-time scoring list, behind only **Pete Maravich**, Freeman Williams, and Lionel . . . | |
| $\mathbf{d_2}$ | Title: Buzzy Wilkinson 
 **Buzzy Wilkinson** Richard Warren "Buzzy" Wilkinson (November 18, 1932 – January 15, 2016) was an American basketball player who was selected by the Boston Celtics in . . . | |
| $\mathbf{d_3}$ | Title: Gary Hill (basketball) 
 . . . becoming one of four on that team who averaged double figures in points. Senior **Larry Jones** was OCU's leading scorer at 19.7 points a game, sophomore Bud Koper . . . | |
| $\mathbf{q_4}$ | '$\mathbf{q_3}$' +(contents:"maravich") | [**score:** 0.784] |
| $\mathbf{d_1}$ | Title: Alphonso Ford 
 . . . seasons. With 3,165 career points scored in the NCAA Division I, he is 4th on the all-time scoring list, behind only **Pete Maravich**, Freeman Williams, and Lionel . . . | |
| $\mathbf{d_2}$ | Title: Pete Maravich 
 . . . had posted a 3–20 record in the season prior to his arrival. **Pete Maravich** finished his college career in the 1970 National Invitation Tournament, where LSU finished fourth . . . | |
| $\mathbf{d_3}$ | Title: 1970 National Invitation Tournament 
 . . . represented the final college games for LSU great **Pete Maravich**, the NCAA's all-time leading scorer. Maravich finished his three-year career with 3,667 points . . . | |

The previous point extends to the agents' architecture. It is reasonable to hypothesise that the superior performance of T5 is due to two main factors. T5s are bigger models, trained on more data, and have a more expressive prediction process based on encoder-decoders. In addition, they are finetuned on a self-supervised tasks which provides significant headroom. While T5-like LMs seem the obvious choice forward there are open questions concerning exploration. It is not clear how much the model can generalize being trained offline and never being exposed to its own prediction. This moves the learning problem back towards RL. We plan to investigate approaches like Decision Transformers (Chen et al., 2021) next, as a natural framework for expressing the search task within LMs, while at the same time bringing back in key RL concepts such as expected returns and learning also from negative experiences, possibly produced by different sources; e.g., MuZero.

We would like to note that pre-trained language models of the kind used here have been shown to capture societal biases (Tan & Celis, 2019; Webster et al., 2020), which motivates a broad discussion about potential harms and mitigations (Blodgett et al., 2020; Bender et al., 2021). We have no reason to believe our architectures would exacerbate biases, but the overall problems may persist. We also hope that end-to-end optimization methods based on composite rewards, as in this proposal, can contribute to solve some of these challenges; e.g., by providing means of adversarial testing, and by including relevant metrics directly in the objective design.

While our agents yield performance comparable to neural retrievers they rely solely on interpretable, symbolic retrieval operations. Exploration and transparency are core objectives of our framework.

## 6 RELATED WORK

Query optimization is an established topic in IR. Methods range from hand-crafted rules (Lawrence & Giles, 1998) to data-driven transformation patterns (Agichtein et al., 2001). Narasimhan et al. (2016) use RL to query the web for information extraction. Nogueira & Cho (2017) and Buck et al. (2018) use RL-trained agents to seek good answers by reformulating questions with seq2seq models. These methods are limited to one-step episodes and queries to plain natural language. This type of modeling is closely related to the use of RL for neural machine translation, whose robustness is currently debated (Choshen et al., 2020; Kiegeland & Kreutzer, 2021). Montazeralghaem et al. (2020) propose a feature-based network to score potential relevance feedback terms to expand a query. Das et al. (2019) propose to perform query reformulation in embedding (continuous) space and find that it can outperform the sequence-based approach. Xiong et al. (2021) successfully use relevance feedback by jointly encoding the question and the text of its retrieved results for multi-hop QA. Other work at the intersection of IR and RL concerns bandit methods for news recommendation (Li et al., 2010) and learning to rank (Yue & Joachims, 2009). Recently, interest in Deep RL for IR has grown (Zhang et al., 2021). There, the search engine is the agent, and the user the environment. In contrast, we view the search problem from the user perspective and thus consider the search engine as the environment.

The literature on searchers' behavior is vast, see e.g. Strzelecki (2020) for an overview of eye-tracking studies. While behavior evolves with interfaces, users keep parsing results fast and frugally, attending to just a few items. From a similar angle, Yuan et al. (2020) offer promising findings on training QA agents with RL for template-based information gathering and answering actions. Most of the work in language-related RL is otherwise centered on synthetic navigation/arcade environments (Hu et al., 2019). This line of research shows that RL for text reading can help transfer (Narasimhan et al., 2018) and generalization (Zhong et al., 2020) in synthetic tasks but skirts the challenges of more realistic language-based problems. On the topic of grammars, Neu & Szepesvári (2009) show that Inverse RL can learn parsing algorithms in combination with PCFGs (Salomaa, 1969).

Current work in OpenQA focuses on the search engine side of the task, typically using dense neural passage retrievers based on a dual encoder framework instead of BM25 (Lee et al., 2019; Karpukhin et al., 2020). Leveraging large pre-trained language models to encode the query and the paragraphs separately led to a performance boost across multiple datasets, not just in the retrieval metrics but also in exact-match score. While Karpukhin et al. (2020) use an extractive reader on the top-k returned paragraphs, Lewis et al. (2020b) further improves using a generative reader (BART (Lewis et al., 2020a)). This design combines the strengths of a parametric memory – the pre-trained LM – with a non-parametric memory – the retrieved Wikipedia passages supplied into the reader's context. This idea of combining a dense retriever with a generative reader is further refined in Izacard & Grave (2021), which fuses multiple documents in the decoding step. A recent line of work is concerned with constraining the model in terms of the number of parameters or retrieval corpus size while remaining close to state-of-the-art performance (Min et al., 2021). This effort led to a synthetic dataset of 65 million *probably asked* questions (Lewis et al., 2021) used to do a nearest neighbor search on the question – no learned parameters needed – or train a closed-book generative model.

## 7 CONCLUSION

Learning to search sets an aspiring goal for AI, touching on key challenges in NLU and ML, with far reaching consequences for making the world's knowledge more accessible. Our paper provides the following contributions. First, we open up the area of search session research to supervised language modeling. Second, we provide evidence for the ability of RL to discover successful search policies in a task characterized by multi-step episodes, sparse rewards and a high-dimensional, compositional action space. Lastly, we show how the search process can be modeled via transparent, interpretable machine actions that build on principled and well-established results in IR and NLU.

Our findings seem to agree with a long-standing tradition in psychology that argues against radical behaviorism – i.e., pure reinforcement-driven learning, from *tabula rasa* – for language (Chomsky, 1959). RL agents require a remarkable share of hard-wired domain knowledge. LM-based agents are easier to put to use, but they need abundant task-specific data for fine tuning. Supplied with the right inductive bias, LM and RL search agents prove surprisingly effective. Different architectures learn different policies, suggesting broad possibilities in the design space for future work.

**Reproducibility statement**   The code for this paper has already been open sourced (reference omitted for anonymity) and included with the supplementary material. We will make sure results are fully reproducible by updating the open source code based on the final version of the paper after completion of the review process.

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

APPENDIX

A   ROCCHIO QUERY EXPANSIONS

Algorithm 1 provides a schematic summary of the procedure for generating Rocchio sessions. We omit the terms source for simplicity and readability, but it should be straightforward to reconstruct. Table A.7 shows an example of a session produced by Rocchio expansions using the full grammar.

---

**Algorithm 1:** Rocchio Query Expansions

**input** : A question-answer pair $(q, a)$, $\texttt{k} = 5$, $\texttt{num\_steps} = 20$, $\texttt{N} = 100$, $\texttt{M} = 100$
**output**: A set of observation-query expansion pairs for training a T5 agent $\texttt{RQE} = \{(o_t, \Delta q_t)\}$

$\texttt{RQE} \leftarrow \emptyset \ q_t \leftarrow q$;
$\mathcal{D}_t \leftarrow \emptyset$ ;                                   // Unique documents found in the session
$q_* \leftarrow q + (\text{contents:}"a")$ ;                               // The ideal query
$\mathcal{D}_* \leftarrow \texttt{LuceneBM25}(q_*)$ ;          // Use search to get the top $k$ documents
// Use the agent PS and MR components to rerank the documents, extract
   answer spans to compute the snippets from the top $k$ results, and
   compile the observation (cf.  also Appendix B)
$o_* \leftarrow \texttt{ComputeObservation}(q, q_*, \mathcal{D}_*, k)$;
$\Sigma_* \leftarrow \texttt{ComputeDictionary}(o_*)$ ;              // Collect good search terms
**for** $t \leftarrow 1$ **to** $\texttt{num\_steps}$ **do**
  |  $\mathcal{D}_t \leftarrow \mathcal{D}_t \cup \texttt{LuceneBM25}(q_t)$;
  |  $o_t \leftarrow \texttt{ComputeObservation}(q, q_t, \mathcal{D}_t, k)$;
  |  $\Sigma_t \leftarrow \texttt{ComputeDictionary}(o_t)$;
  |  $\Sigma^\uparrow \leftarrow \Sigma_* \cap \Sigma_t, \Sigma^\downarrow \leftarrow \Sigma_t - \Sigma_*$;
  |  $s_t \leftarrow \texttt{ComputeScore}(q, \mathcal{D}_t, k)$;          // Compute the score using Eq.(7)
  |  $\texttt{max\_score} \leftarrow s_t$;
  |  $\texttt{best\_action} \leftarrow \emptyset, \texttt{num\_tries} \leftarrow 0$;
  |  // Evaluate all available operators/fields, and also without
  |  **for** $\texttt{op}, \texttt{field} \in \{\{+, -, \wedge 0.1, \wedge 2, \wedge 4 \wedge 6 \wedge 8\} \times \{\text{contents:}, \text{title:}\}\} \cup \{('','')\}$ **do**
  |  |  **for** $w \in \texttt{TopNTermsByLuceneIDF}(\Sigma_t, N)$ **do**
  |  |  |  **if** $(\texttt{op} == '-' \wedge w \in \Sigma^\downarrow) \vee (\texttt{op} \neq '-' \wedge w \in \Sigma^\uparrow) \wedge (\texttt{num\_tries} < M)$ **then**
  |  |  |  |  $\Delta q_t \leftarrow \texttt{op}(\texttt{field:}'w')$;
  |  |  |  |  $q' \leftarrow q_t + \Delta q_t$;
  |  |  |  |  $\mathcal{D}' \leftarrow \mathcal{D}_t \cup \texttt{LuceneBM25}(q')$;
  |  |  |  |  $s' \leftarrow \texttt{ComputeScore}(q, \mathcal{D}', k)$;
  |  |  |  |  $\texttt{num\_tries} \leftarrow \texttt{num\_tries} + 1$;
  |  |  |  |  **if** $s' > \texttt{max\_score}$ **then**
  |  |  |  |  |  $\texttt{max\_score} \leftarrow s'$;
  |  |  |  |  |  $\texttt{best\_action} \leftarrow \Delta q_t$;
  |  |  |  |  **end**
  |  |  |  **else**
  |  |  |  |  continue;
  |  |  |  **end**
  |  |  **end**
  |  **end**
  |  **if** $\texttt{max\_score} > s_t$ **then**
  |  |  // If the best action improves the score, add this step to the
  |  |     data, and continue the session
  |  |  $q_t \leftarrow q_t + \texttt{best\_action}$;
  |  |  $\texttt{RQE} \leftarrow \texttt{RQE} \cup (o_t, \texttt{best\_action})$;
  |  **else**
  |  |  **return** $\texttt{RQE}$;
  |  **end**
**end**
**return** $\texttt{RQE}$

---

## B OBSERVATION BUILDING DETAILS

This section provides more details and examples about the encoding of observations for both the MuZero and the T5 agent. As described in Section 2.1, the main part of the observation consists of the top-5 documents from all results retrieved so far, $\cup_{i=0}^{t} \mathcal{D}_i$. The documents are sorted according to the PS score and reduced in size by extracting fixed-length snippets around the DPR reader's predicted answer. Moreover, the corresponding Wikipedia article title is appended to each document snippet. In addition to the top documents, the observation includes the original question and information about any previous refinements. While the main part of the observation is shared between the MuZero and the T5 agent, there are differences in the exact representation. The following two paragraphs give a detailed explanation and example for both agents.

**MuZero Agent's State (cf. §2.1)**  The MuZero agent uses a custom BERT (initialized from BERT-base) with additional embedding layers to represent the different parts of the observation. It consists of four individual embedding layers as depicted in Figure A.1. At first, the standard layer for the tokens of the query, the current tree, and the current top documents $D^5$. The second layer assigns a type ID to each of the tokens representing if a token is part of the query, the tree, the predicted answer, the context, or the title of a document. The last two layers add scoring information about the tokens as float values. We encode both the inverse document frequency (IDF) of a word and the documents' passage selection (PS) score. Figure A.1 shows a concrete example of a state used by the MuZero agent.

| Layer | Query | Tree | Document Results |
|---|---|---|---|
| Tokens | $q_0$ | $l_0, \ldots, l_m$ | $a_0, c_0, t_0, \ldots, a_n, c_n, t_n$ |
| Type | $\mathrm{ID}_q$ | $\mathrm{ID}_{\mathrm{tree}}$ | $\mathrm{ID}_a, \mathrm{ID}_c, \mathrm{ID}_t, \ldots, \mathrm{ID}_a, \mathrm{ID}_c, \mathrm{ID}_t$ |
| IDF Score | $\mathrm{idf}(q_0)$ | $\mathrm{idf}(l_0), \ldots, \mathrm{idf}(l_m)$ | $\mathrm{idf}(a_0), \mathrm{idf}(c_0), \mathrm{idf}(t_0), \ldots, \mathrm{idf}(a_n), \mathrm{idf}(c_n), \mathrm{idf}(t_n)$ |
| PS Score | $0$ | $0$ | $\mathrm{PS}(d_0), \ldots, \mathrm{PS}(d_n)$ |

Figure A.1: Schematic illustration of the MuZero search agent's state for the BERT representation function.

Table A.1: Example state of the MuZero search agent that is the input to the BERT representation function. The 'Type' layer encodes the state part information for each token. The 'IDF' and 'PS' layer are additional layers with float values of the IDF and the PS score of the input tokens, respectively.

| Tokens | [CLS] | who | carries | the | burden | of | going | forward | with | evidence | in | a | trial | |
|---|---|---|---|---|---|---|---|---|---|---|---|---|---|---|
| **Type** | [CLS] | query | query | query | query | query | query | query | query | query | query | query | query | ... |
| **IDF** | 0.00 | 0.00 | 6.77 | 0.00 | 7.77 | 0.00 | 5.13 | 5.53 | 0.00 | 5.28 | 0.00 | 0.00 | 5.77 | |
| **PS** | 0.00 | 0.00 | 0.00 | 0.00 | 0.00 | 0.00 | 0.00 | 0.00 | 0.00 | 0.00 | 0.00 | 0.00 | 0.00 | |

| Tokens | [SEP] | [pos] | [content] | burden | ##s | [neg] | [title] | sometimes | [SEP] | lit | ##igan | ##ts | [SEP] | |
|---|---|---|---|---|---|---|---|---|---|---|---|---|---|---|
| **Type** | [SEP] | tree | tree | tree | tree | tree | tree | tree | [SEP] | answer | answer | answer | [SEP] | ... |
| **IDF** | 0.00 | 0.00 | 0.00 | 9.64 | 9.64 | 0.00 | 0.00 | 4.92 | 0.00 | 10.64 | 10.64 | 10.64 | 0.00 | |
| **PS** | 0.00 | 0.00 | 0.00 | 0.00 | 0.00 | 0.00 | 0.00 | 0.00 | 0.00 | -3.80 | -3.80 | -3.80 | -3.80 | |

| Tokens | kinds | for | each | party | , | in | different | phases | of | litigation | . | the | burden | |
|---|---|---|---|---|---|---|---|---|---|---|---|---|---|---|
| **Type** | context | context | context | context | context | context | context | context | context | context | context | context | context | ... |
| **IDF** | 7.10 | 0.00 | 0.00 | 4.36 | 17.41 | 0.00 | 4.18 | 7.46 | 0.00 | 7.92 | 17.41 | 0.00 | 7.77 | |
| **PS** | -3.80 | -3.80 | -3.80 | -3.80 | -3.80 | -3.80 | -3.80 | -3.80 | -3.80 | -3.80 | -3.80 | -3.80 | -3.80 | |

| Tokens | suspicion | " | , | " | probable | cause | " | ( | as | for | [SEP] | evidence | [SEP] | |
|---|---|---|---|---|---|---|---|---|---|---|---|---|---|---|
| **Type** | context | context | context | context | context | context | context | context | context | context | [SEP] | title | [SEP] | ... |
| **IDF** | 7.80 | 17.41 | 17.41 | 17.41 | 7.91 | 5.41 | 17.41 | 17.41 | 0.00 | 0.00 | 0.00 | 5.28 | 0.00 | |
| **PS** | -12.20 | -12.20 | -12.20 | -12.20 | -12.20 | -12.20 | -12.20 | -12.20 | -12.20 | -12.20 | -12.20 | -12.20 | -12.20 | |

**T5 Agent's State (cf. §2.3)**  T5 represents the state as a flat string. The input is a concatenation of the original query, zero or more expansions, and five results. For each result, we include the answer given by the reader, the document's title, and a span centered around the answer. The prediction target is simply the next expansion. See Table A.2 for a full example.

Table A.2: Example state (input) and prediction (target) of the T5 agent with linebreaks and emphasis added for readability. We use a 30 token span in our experiments.

| | |
|---|---|
| Input | Query: 'how many parts does chronicles of narnia have'.
Contents must contain: lewis.
Contents cannot contain: battle boost 2.0.

Answer: 'seven'.
Title: 'The Chronicles of Narnia'.
Result: The Chronicles of Narnia is a series of *seven* fantasy novels by C. S. Lewis. It is considered a classic of children's literature and is the author's best-known work, having...

Answer: 'seven'.
Title: 'The Chronicles of Narnia (film series)'.
Result: '"The Chronicles of Narnia", a series of novels by C. S. Lewis. From the *seven* books, there have been three film adaptations so far – (2005), "" (2008) and "" (2010)...

Answer: 'seven'.
Title: 'Religion in The Chronicles of Narnia'.
Result: 'Religion in The Chronicles of Narnia "The Chronicles of Narnia" is a series of *seven* fantasy novels for children written by C. S. Lewis. It is considered a classic of...

Answer: 'seven'.
Title: 'The Chronicles of Narnia'.
Result: 'Lewis's early life has parallels with "The Chronicles of Narnia". At the age of *seven* , he moved with his family to a large house on the edge of Belfast...

Answer: 'Two'.
Title: 'The Chronicles of Narnia'.
Result: 'found in the most recent HarperCollins 2006 hardcover edition of "The Chronicles of Narnia". *Two* other maps were produced as a result of the popularity of the 2005 film ... |
| Target | Contents must contain: novels |

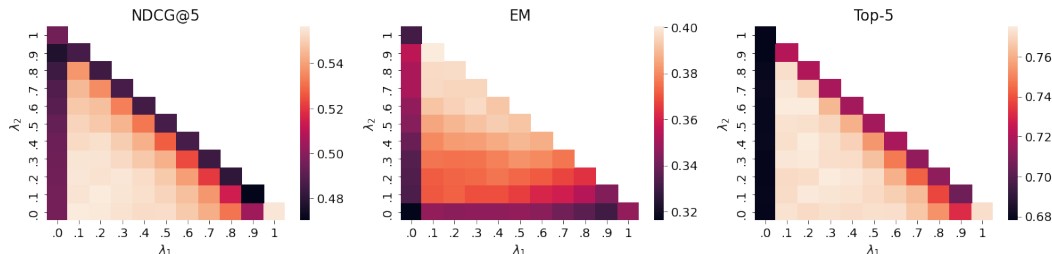

Figure A.2: Performance (NDCG@5, EM, and Top-5, respectively) of the Rocchio episodes from NQ-dev guided by the composite score of Equation 7, as a function of the coefficients $\lambda_1$ and $\lambda_2$.

## C  REWARD DETAILS

We investigate the effects of the three score components in the definition of the composite scoring function of Eq. 7. As mentioned in Section 4, in our experiments we have observed that using only the $\text{NDCG}_k$ score as reward signal (i.e., setting $\lambda_1 = \lambda_2 = 0$ in Eq. 7) has several limitations. This motivated us to introduce the $\text{NDCEM}_k$ and PS components with the intent of: 1) providing further guidance to the agent (whenever $\text{NDCG}_k$ cannot be increased, the agent can further refine the query by increasing $\text{NDCEM}_k$ or PS), and 2) regularizing the search episodes by making the score more robust with respect to exploratory behaviors that could lead to drift.

We run a grid search over the reward coefficients $\lambda_1, \lambda_2$ and, for each of their values, we evaluate the performance of the Rocchio query expansions on NQ Dev (for a high throughput, we select grammar **G3** and set $N = M = 20$). Figure A.2 shows the respective end-to-end performance in terms of our three main quality metrics: NDCG@5, EM, and Top-5.

The results in Figure A.2 support our intents: by introducing $\text{NDCEM}_k$ and PS scores in the reward (i.e., setting $\lambda_1, \lambda_2 > 0$), the Rocchio expansions can achieve significantly higher performance, in all the three metrics, with respect to using only an $\text{NDCG}_k$ score ($\lambda_1 = \lambda_2 = 0$) (notably, it improves also the NDCG@5 itself, meaning that the agent is not trading-off performance metrics but it is indeed producing higher quality sessions). It is also worth pointing out the role of the $\text{NDCEM}_k$ score component, weighted by coefficient $\lambda_2$. Notice that good NDCG@5 and Top-5 performance could be achieved also setting $\lambda_2 = 0$ (see, e.g., the bottom-right corner $\lambda_1 = 1, \lambda_2 = 0$). However, this leads to definitely worse EM results compared to when $\lambda_2 > 0$. Intuitively, a $\text{NDCEM}_k$ component $\lambda_2 > 0$ ensures that the returned documents, in addition to containing the gold answer (thus having high NDCG@5 and Top-5), are also relevant for the query (thus reaching a high EM). Hence, it is crucial to prevent semantic drifts. Based on these results we set $\lambda_1 = 0.2, \lambda_2 = 0.6$, which is a sweet spot in Figure A.2.

## D  MODEL AND TRAINING CONFIGURATION

**MuZero**  The MuZero agent learner, which performs both inference and training, runs on a Cloud TPU v2 with 8 cores which is roughly equivalent to 10 Nvidia P100 GPUs in terms of TFLOPS.[6] One core is allocated for training and 7 cores are allocated for inference. We use 500 CPU based actors along with 80 actors dedicated to evaluation. Each agent is trained for 1.6 million steps, with 100 simulations per step, at an approximate speed of 10,000 steps per hour. In total, training takes about 10 days. Hyperparameters are listed in Table A.3.

Table A.3: Hyperparameters for MuZero.

| Parameter | Value |
|---|---|
| Simulations per Step | 100 |
| Actor Instances | 500 |
| Training TPU Cores | 1 |
| Inference TPU Cores | 7 |
| Initial Inference Batch Size ($h_\theta$) | 4 per core |
| Recurrent Inference Batch Size ($f_\theta, g_\theta$) | 32 per core |
| LSTM Units ($g_\theta$) | One layer of 512 |
| Feed-forward Units ($f_\theta$) | One layer of 32 |
| Training Batch Size | 16 |
| Optimizer | SGD |
| Learning Rate | 1e−4 |
| Weight Decay | 1e−5 |
| Discount Factor ($\gamma$) | .9 |
| Unroll Length ($K$) | 5 |
| Max. #actions Expanded per Step | 100 |
| Max. context tokens from document title | 10 |
| Max. context tokens from document content | 70 |

**T5**  The T5 agent is trained on 16 Cloud TPU v3, starting from the pre-trained T5-11B checkpoint. We mostly follow the suggested hyperparameter choices given by Raffel et al. (2020), see Table A.4. We train for about 2 days at roughly 3 steps per second for a total of 50k steps and select a checkpoint based on dev performance.

---

[6]The Cloud TPU v2 has a peak performance of 180 TFLOPS (https://cloud.google.com/tpu), whereas the Nvidia P100 GPU goes up to 18.7 TFLOPS depending on precision (https://cloud.google.com/compute/docs/gpus).

Table A.4: Hyperparameters for T5.

| Parameter | Value |
|---|---|
| Number of Parameters | 11B |
| Encoder/Decoder Layers | 24 |
| Feed-forward dimension | 65536 |
| KV dimension | 128 |
| Model dimension | 1024 |
| Number of Heads | 128 |
| Batch Size (in tokens) | 65536 |
| Dropout Rate | 0.1 |
| Learning Rate (constant) | 0.0005 |
| Optimizer | AdaFactor |
| Maximum input length (tokens) | 512 |
| Maximum target length (tokens) | 32 |
| Finetuning steps on NQ Train | 41400 |
| Max. context tokens from document title | 10 |
| Max. context tokens from document content | 30 |

## E  RESULTS

Table A.5 reports the results for the different versions of the T5 agent, evaluated on dev. We don't evaluate all agents with the generative answer system, for answer quality we report only the performance of the internal machine reader (EM-MR). Table A.6 reports extended results, including for NQ Dev and the PS/MR component answer quality eval (EM-MR).

Table A.5: Results of all T5 Agents on NQ Dev.

| Version | NDCG@5 | Top-1 | Top-5 | EM-MR | Reward |
|---|---|---|---|---|---|
| G0 | 40.75 | 52.12 | 64.93 | 30.22 | 33.30 |
| G1 | 43.10 | 52.12 | 66.09 | 29.50 | 35.55 |
| G2 | 41.16 | 51.51 | 63.54 | 30.03 | 33.81 |
| G3 | 41.69 | 51.34 | 64.17 | 29.77 | 33.95 |
| G4 | 41.53 | 50.98 | 63.49 | 29.70 | 34.25 |

Table A.6: Results on NQ Dev and Test.

| Metric | Data | BM25 | PS($\mathcal{D}_0$) | MZ | T5-G1 | MZ+T5s | DPR | RQE-G4 |
|---|---|---|---|---|---|---|---|---|
| NDCG@5 | Dev | 19.83 | 22.95 | 30.76 | 43.10 | 45.30 | - | 64.89 |
|  | Test | 21.51 | 24.82 | 32.23 | 44.27 | 46.22 | - | 65.24 |
| Top-1 | Dev | 28.17 | 43.06 | 46.02 | 52.12 | 54.15 | - | 74.99 |
|  | Test | 28.67 | 44.93 | 47.97 | 52.60 | 54.29 | 52.47 | 73.74 |
| Top-5 | Dev | 50.47 | 50.47 | 57.71 | 66.09 | 70.05 | - | 88.21 |
|  | Test | 53.76 | 53.76 | 59.97 | 66.59 | 71.05 | 72.24 | 88.17 |
| EM-MR | Dev | 15.31 | 25.15 | 27.17 | 29.50 | 31.12 | - | 47.38 |
|  | Test | 14.79 | 25.87 | 28.19 | 30.08 | 30.58 | 41.50 | 46.34 |
| EM-T5 | Dev | 28.98 | 40.70 | 32.48 | 44.75 | 44.47 | - | 63.78 |
|  | Test | 28.78 | 41.14 | 32.60 | 44.04 | 44.35 | 41.50 | 62.35 |

## F  DETAILS AND EXAMPLES FOR THE GRAMMAR-GUIDED MCTS

$$
\begin{aligned}
\text{Q} &\Rightarrow \text{W Q} \mid \text{U Q} \mid \text{STOP} \\
\text{U} &\Rightarrow \text{Op Field W} \\
\text{Op} &\Rightarrow - \mid + \mid \wedge_i \\
&\quad i \in \{0.1, 2, 4, 6, 8\} \\
\text{Field} &\Rightarrow \text{title} \mid \text{contents} \\
\text{W}^\text{x} &\Rightarrow \text{V}^\text{x} \mid \text{V}^\text{x}\, \overline{\text{W}}^\text{x} \\
\overline{\text{W}}^\text{x} &\Rightarrow \overline{\text{V}}^\text{x} \mid \overline{\text{V}}^\text{x}\, \overline{\text{W}}^\text{x} \\
V^x &\Rightarrow \{w \mid w \in V_\text{B} \wedge \\
&\quad \text{trie}(\Sigma^\text{x}).\text{HasSubstring}(w)\} \\
\overline{V}^x &\Rightarrow \{w \mid \#w \in V_\text{B}^\# \wedge \\
&\quad \text{trie}(\Sigma^\text{x}).\text{HasSubstring}(\overrightarrow{w})\}
\end{aligned}
$$

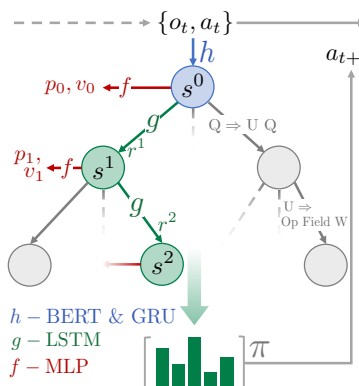

(a) The productions of the query grammar: x identifies a specific vocabulary induced by the aggregated results at time $t$ (index omitted), $V_\text{B}$ ($V_\text{B}^\#$) is the BERT wordpiece prefix (suffix) vocabulary, $\overrightarrow{w}$ denotes the string ending at $w$, including the preceding wordpieces.

(b) The MuZero MCTS with grammar-guided node expansions represented as edge labelled with CFG rules.

Figure A.3

Figure A.3a lists the detailed rules schemata for the query grammar used by the MuZero agent – explained in Section 3.1. An optional STOP rule allows the agent to terminate an episode and return the results collected up to that point. Using the BERT sub-word tokens as vocabulary allows us to generate a large number of words with a total vocabulary size of $\sim$30k tokens.

Our implementation of MuZero modifies the MCTS to use the query grammar for efficient exploration. Figure A.3b shows the different network components used during MCTS. Each node expansion is associated with a grammar rule. When the simulation phase is complete, the visit counts collected at the children of the MCTS root node provide the policy $\pi$ from which the next action $a_{t+1}$ is sampled.

Each simulation corresponds to one or more hypothetical follow-up queries (or fragments) resulting from the execution of grammar rules. The MCTS procedure executes Depth-First node expansions, guided by the grammar, to generate a query top-down, left-to-right, in a forward pass. To control the process, we add two data structures to MCTS nodes: a stack $\gamma$, and an output buffer $\omega$: $\gamma$ contains a list of unfinished non-terminals, $\omega$ stores the new expansion. The stack is initialized with the start symbol $\gamma = [Q]$. The output buffer is reset, $\omega = []$, after each document search. When expanding a node, the non-terminal symbol on the top of $\gamma$ is popped, providing the left-hand side of the rule associated with the new edge. Then, symbols on the right-hand side of the rule are pushed right-to-left onto $\gamma$. When a terminal rule is applied, the terminal symbol is added to $\omega$. The next time $\gamma$ contains only Q, $\omega$ holds the new query expansion term $\Delta q_t$ to be appended to the previous query $q_t$ for search.

Figure A.4 illustrates the process. Nodes represent the stack $\gamma$ and output buffer $\omega$. Edges are annotated with the rule used to expand the node. We illustrate the left-branching expansion. Starting from the top, the symbol "Q" is popped from the stack, and a compatible rule, "Q $\Rightarrow$ W Q", is sampled. The symbols "W" and "Q" are added to the stack for later processing. The agent expands the next node choosing to use the document content vocabulary (W $\Rightarrow$ W$^\beta$), then it selects a vocabulary prefix ('dial'), adding it to the output buffer $\omega$, followed by a vocabulary suffix ('ects'). At that point, the stack contains only Q, and the content of $\omega$ contains a new expansion, the term 'dialects'. A latent search step is simulated through MuZero's $g_\theta$ sub-network. Then the output buffer $\omega$ is reset.

After the latent search step, the simulation is forced to use the full trie (W $\Rightarrow$ W$^\text{idx}$), which includes all terms in the Lucene index. This is necessary since there is no observable context that can be used to restrict the vocabulary. Instead of simply adding an OR term (Q $\Rightarrow$ W Q), the right branch of the example selects an expansion with unary operator and field information (Q $\Rightarrow$ U Q).

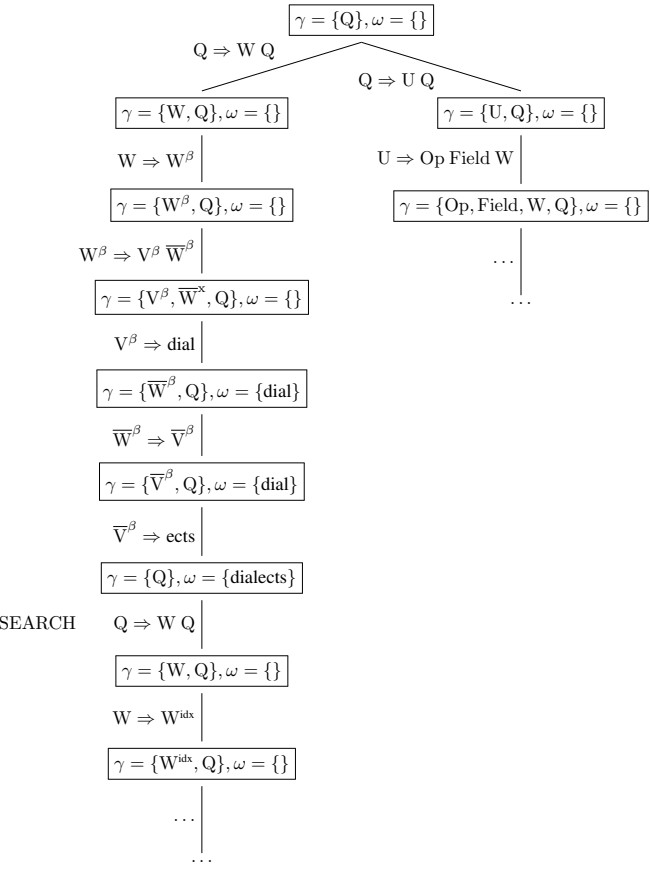

Figure A.4

## G  SEARCH SESSION EXAMPLES

Table A.7 shows an example of a session produced by Rocchio expansions using the full grammar.
Table A.8 shows a session generated by the MuZero agent.

Table A.7: Example of a Rocchio Query Expansion session.

| | | |
|---|---|---|
| $q_0$ | who were the judges on the x factor | [**score:** 0.043] |

    $d_1$    Title: The X Factor (Australian TV series)
        ...Chloe Maxwell. After "The X Factor" was revived for a second season in 2010, **Natalie Garonzi** became the new host of "The Xtra Factor" on the Seven Network's digital channel ...

    $d_2$    Title: X Factor (Icelandic TV series)
        ...the show. The judges were the talent agent and businessman Einar Bárðarson, rock musician **Elínborg Halldórsdóttir** and pop singer Paul Oscar. Previously both Einar and Páll Óskar had been judges ...

| | | |
|---|---|---|
| $q_1$ | '$q_0$' (contents:"confirmed"∧4) | [**score:** 0.551] |

    $d_1$    Title: The X Factor (U.S. season 2)
        ...Khloé Kardashian and "Extra" host Mario Lopez, replacing Steve Jones from the previous year. **Simon Cowell** and L.A. Reid returned as judges, while Paula Abdul and Nicole Scherzinger were replaced ...

    $d_2$    Title: The X Factor (New Zealand series 1)
        ...The series screened on Sunday and Monday evenings. "The X Factor" was created by **Simon Cowell** in the United Kingdom and the New Zealand version is based on the original ...

| | | |
|---|---|---|
| $q_2$ | '$q_1$' (title:"2"∧8) | [**score:** 0.678] |

    $d_4$    Title: The X Factor (U.S. season 2)
        ...It was also reported that Cowell was in talks with **Britney Spears** for her to join the show, reportedly offering her $15 million. On May 9, reports surfaced that Spears ...

    $d_5$    Title: H.F.M. 2 (The Hunger for More 2)
        ...Banks' confirmed that Cardiak produces four songs on the album. Confirmed guests include **Eminem, Kanye West**, Lloyd, Juelz Santana, 50 Cent, Styles P, Raekwon, Tony Yayo, Jeremih and ...

| | | |
|---|---|---|
| $q_3$ | '$q_2$' +(contents:"britney") | [**score:** 0.804] |

    $d_3$    Title: The X Factor (U.S. TV series)
        ...Reid, former "The X Factor" judge Cheryl Cole, and Cowell's former "American Idol" colleague **Paula Abdul** were confirmed to join Cowell in the judging panel. Cole was dismissed from the ...

    $d_4$    Title: The X Factor (New Zealand series 2)
        ...from the first series, **Stan Walker and Melanie Blatt, were confirmed to return as judges. The other two judges from the first series, Daniel Bedingfield** and Ruby Frost, were ...

    $d_5$    Title: The X Factor (U.S. season 2)
        ...It was also reported that Cowell was in talks with **Britney Spears** for her to join the show, reportedly offering her $15 million. On May 9, reports surfaced that Spears ...

| | | |
|---|---|---|
| $q_4$ | '$q_3$' (contents:"cowell"∧4) | [**score:** 0.926] |

    $d_1$    Title: The X Factor (U.S. season 2)     (BM25 Rank: 15)
        ...Khloé Kardashian and "Extra" host Mario Lopez, replacing Steve Jones from the previous year. **Simon Cowell** and L.A. Reid returned as judges, while Paula Abdul and Nicole Scherzinger were replaced ...

    $d_2$    Title: The X Factor (New Zealand series 1)     (BM25 Rank: 195)
        ...The series screened on Sunday and Monday evenings. "The X Factor" was created by **Simon Cowell** in the United Kingdom and the New Zealand version is based on the original ...

    $d_3$    Title: Louis Walsh     (BM25 Rank: >1000)
        ...from bronchitis. He joined the other season two judges: L.A. Reid, Demi Lovato and **Britney Spears**, and was introduced with the line, "When Simon needs an opinion from someone ...

    $d_4$    Title: The X Factor (U.S. TV series)     (BM25 Rank: 206)
        ...Reid, former "The X Factor" judge Cheryl Cole, and Cowell's former "American Idol" colleague **Paula Abdul** were confirmed to join Cowell in the judging panel. Cole was dismissed from the ...

    $d_5$    Title: Simon Cowell     (BM25 Rank: >1000)
        ...the winning judge. Cowell and Reid returned for season 2, while Demi Lovato and **Britney Spears** joined the judging panel as replacements for Abdul and Scherzinger. This season launched another ...

Table A.8: Example of a MuZero agent search session.

| | | | |
|---|---|---|---|
| $q_0$ | who won the wwe money in the bank | | [**score:** 0.071] |

$d_1$    Title: Money in the Bank (2017)
. . . the pre-show and two Money in the Bank ladder matches on the main card. In the main event, Baron Corbin won the men's ladder match, earning a contract for a WWE Championship match, while **Carmella** controversially won the first women's ladder match to earn a SmackDown Women's Championship match contract. In another prominent match, Jinder Mahal defeated Randy Orton to retain the WWE Championship. The event was also notable . . .

| | | |
|---|---|---|
| $q_1$ | '$q_0$' (contents:"jinder"∧2) | [**score:** 0.130] |

$d_1$    Title: Money in the Bank ladder match
. . . For the traditional ladder match, which had a contract for a match for SmackDown's WWE Championship, SmackDown Commissioner Shane McMahon announced AJ Styles, Shinsuke Nakamura, Dolph Ziggler, Sami Zayn, and **Baron Corbin** as the original five participants. United States Champion Kevin Owens was added after he convinced Shane to make him the sixth participant. Mojo Rawley had the opportunity to make it a seven-man match if he could defeat WWE . . .

| | | |
|---|---|---|
| $q_2$ | '$q_1$' (contents:"dolph"∧2) | [**score:** 0.138] |
| $q_3$ | '$q_2$' (contents:"won"∧2) | [**score:** 0.138] |
| $q_4$ | '$q_3$' (contents:"zayn"∧2) | [**score:** 0.414] |

$d_1$    Title: Money in the Bank (2018)
. . . April 23 episode of "Raw", Bobby Lashley performed a one-handed suspended vertical suplex on Sami Zayn, which Zayn claimed gave him vertigo and why he missed the Greatest Royal Rumble. At Backlash, Lashley and **Braun Strowman** defeated Kevin Owens and Sami Zayn. During an interview on the May 7 episode, Lashley spoke dearly about his family, including his three sisters. In response, Zayn said that Lashley was not a nice . . .

| | | |
|---|---|---|
| $q_5$ | '$q_4$' (contents:"strowman"∧2) | [**score:** 0.587] |

$d_2$    Title: Kevin Owens
. . . both men would be awarded Raw contracts by Raw Commissioner Stephanie McMahon. Later that night, Owens teaming up with Zayn, The Miz, Curtis Axel and Bo Dallas and lost to Finn Bálor, Seth Rollins, **Braun Strowman**, Bobby Lashley and Bobby Roode in a 10-man tag team match. On the May 7 episode of "Raw", he lost a Money in the Bank qualifying match to Braun Strowman. At Backlash, Owens . . .

| | | |
|---|---|---|
| $q_6$ | '$q_5$' (contents:"first"∧2) | [**score:** 0.587] |
| $q_7$ | '$q_6$' (contents:"roode"∧2) | [**score:** 0.848] |

$d_1$    Title: Bobby Lashley      (BM25 Rank: >1000)
. . . Lashley participated in the Greatest Royal Rumble at the namesake event, entering at #44 and scoring two eliminations, but was eliminated by **Braun Strowman**. The first month of Lashley's return would see him in a number of tag-team matches, teaming with the likes of Braun Strowman, Bobby Roode, Finn Balor, Seth Rollins, and Roman Reigns, all being in victorious efforts against the likes of Kevin Owens, Sami Zayn, Jinder . . .

$d_2$    Title: Kevin Owens and Sami Zayn      (BM25 Rank: >1000)
. . . following week, as she would award both men "Raw" contracts. Later that night, Owens and Zayn teamed up with The Miz and The Miztourage (Curtis Axel and Bo Dallas), against Finn Balor, Seth Rollins, **Braun Strowman**, Bobby Lashley, and another new "Raw" member Bobby Roode, but their team was defeated. The following week on "Raw", they teamed up in an unsuccessful attempt to defeat Strowman and Lashley. On the . . .

$d_3$    Title: Money in the Bank (2018)      (BM25 Rank: 282)
. . . April 23 episode of "Raw", Bobby Lashley performed a one-handed suspended vertical suplex on Sami Zayn, which Zayn claimed gave him vertigo and why he missed the Greatest Royal Rumble. At Backlash, Lashley and **Braun Strowman** defeated Kevin Owens and Sami Zayn. During an interview on the May 7 episode, Lashley spoke dearly about his family, including his three sisters. In response, Zayn said that Lashley was not a nice . . .

