# OpenReview forum: "Boosting Search Engines with Interactive Agents"
_ICLR.cc/2022/Conference — ICLR 2022 Submitted_

### Official Review · Reviewer_13nH · 2021-11-03

**Correctness:** 3
**Technical Novelty And Significance:** 2
**Empirical Novelty And Significance:** 3
**Recommendation:** 3
**Confidence:** 4

**Main Review:**

**Strengths:**
- This work is generally well-written with good coverage of the literature.
- Most components are motivated and explained throughout the text.
- There is also extensive qualitative analysis of the query expansion results.

**Weaknesses:**
- Unfortunately, despite the careful fine-tuning and modifications on the compilation of several components, results seem inferior to a neural passage retrieval model. There is a discussion on why this might be the case, and how this work provides the foundations for work on policy synthesis in search and retrieval tasks. But when comparing a distributed MuZero agent trained on 1.6 million steps (approx. 10 days) and a T5 agent with 11 billion parameters over DPR, it makes sense to also report on the number of parameters (model capacity comparison) and the total training times and the number of training examples (time/sample complexity comparison).
- A comparison with related work on RL for query reformulation, e.g., Nogueira & Cho (2017), Buck et al. (2018) and [1]. Albeit these models have been tested in one-step episodes, there is no explanation on why these cannot be trained in a multistep refinement setting.
- Please also include SOTA passage retrieval models, such as [2].

[1] Nogueira, Rodrigo, Jannis Bulian, and Massimiliano Ciaramita. "Learning to coordinate multiple reinforcement learning agents for diverse query reformulation." arXiv preprint arXiv:1809.10658 (2018).
[2] Qu, Yingqi, et al. "RocketQA: An optimized training approach to dense passage retrieval for open-domain question answering." Proceedings of the 2021 Conference of the North American Chapter of the Association for Computational Linguistics: Human Language Technologies. 2021.

**Questions:**
- Just to confirm, for the Rocchio feedback, the method operates in vector space, i.e., w_t and q_t are vector representations?
- Are results reported over multiple trials?
- A proper ablation study over the rewards would have been useful. To my understanding from Figure A.2, if one sets $\lambda_1 = 1$ and $\lambda_2=0$, better performance can be achieved in terms of NDCG and Top-5, rather than the reverse (i.e.,  $\lambda_1 = 0$ and  $\lambda_2 = 1$). Extracting insights for EM is not so clear. Does this mean that the proposed NDCEM does not really benefit the retrieval performance? Please provide further explanations.

**Minor comments:**
- It appears there is a typo after Eq.(1): "This may add a keyword .... $\sum_{idx}$, the search index vocabulary," ==> "$\sum_{idx}$, where $\sum_{idx}$ is the search index vocabulary"
- Please further explain the part after Eq.(3): "This means it is possible to upgrade accessible terms to exact matches, or weight boosting, if they also occur in the *ideal result set* and to exclude accessible terms if they are not present in the *ideal results*"

**Update after rebuttal:**
Thank you to the authors for their response that clarified some of the paper details. I have read the reviews and responses and my score remains unchanged. The main motivation of transparent and interpretable agent-based query reformulation could be a significant contribution but would be better highlighted with experiments on additional datasets, further discussion on the complexity of the method and qualitative results that show the enhanced interpretability over prior "pure RL" works.
(Minor clarification: Albeit a "contemporaneous" work, surprisingly RocketQA has been on arXiv since October 2020.)

**Summary Of The Paper:**

This work designs an RL (customized BERT MuZero) agent for query reformulation, where the underlying observation is defined as an aggregation of windows of answer spans found with a machine reading module, document titles, query tokens, and refinements. The action space involves adding keywords to the query or form search terms with three operators that constrain the search space. Due to the lack of training data, synthetic search sessions are generated with a pretrained transformer model and imitation learning. The reward is a linear combination of NDCG (well-established IR ranking for non-binary relevance), a revised NDCG-like metric for exact matches (NDCEM) and a top-k Passage Score. Results on an OpenQA dataset are *close* to neural passage retrieval methods.



**Summary Of The Review:**

While this is, as the authors explain, an endeavor that required customization of several components to reach comparable performance with DPR, the method proposed appears to be computationally expensive without significant improvements over neural retrieval. Alongside the lack of comparison over prior work on RL for query reformulation and the lack of discussion over other dimensions apart from performance (and maybe some qualitative analysis), the paper will benefit from further extensions and revisions.

---

> ### Author Response · Authors · 2021-11-18
> **Response to Reviewer 13nH**
>
> __We first address specific weak points brought up by the reviewer:__
>
> The main purpose of the paper was to open up the space of modeling search as a machine-learned controlled process, by means of discrete, interpretable actions. Naturally, we also want the proposal to be competitive, and, obviously, it is. SOTA is a non-goal. Particularly because it is easy to imagine that the best results will come from hybrid approaches. We agree that there is additional complexity, this is intrinsic in the very idea of an iterative, rather than one-shot, method. However, we hope that complexity will be brought down as this family of methods is better understood but we also highlight that transparency and interpretability are worth additional costs.
>
> We are familiar with the work on policy-based reformulation (Nogueira and Cho. 2017, Buck et al. 2018). We found empirically that it is extremely time-consuming to replicate those results due to the instability of policy gradient. We cite this line of work and mention the current debate on the value of policy gradient methods in seq2seq formulations (Choshen et al. 2020, Kiegeland & Kreutzer, 2021).
>
> Previous work differs from our not only through the use of policy gradient vs. MuZero, but also the limitation of the action space and multi step refinement. We found the constrained action space to be critical to reliable training. The agents also achieve relevant gains after the first step, confirming the efficacy of multi-step episodes.
>
> We include a mention of the current SOTA. We note however, that the proposed RocketQA paper falls within the _"contemporaneous"_ work (work published at a peer-reviewed venue on or after June 5) according to the ICLR guidelines.
>
> __Regarding the specific questions:__
>
> The Rocchio procedure is all in language space $q_t$ is a natural language query; e.g., "who scored the most …", the terms $w_t$ available for upgrading (+, boost) and downgrading (-) are represented as strings.
>
> Results reported are single trial runs, based on a complete model parametrization selected a priori on the dev data.
>
> In Appendix C, in the updated draft, we have added a more in-depth discussion regarding the usefulness of the three different reward components in accordance with the results of Fig A.2. The reviewer is correct in that setting $\lambda_1=1$, $\lambda_2=0$ (hence, using only the PS score) leads to better NDCG and Top-5 performance w.r.t. to setting $\lambda_1=0$, $lambda_2=1$. However, from this fact one cannot conclude that NDCEM does not improve the retrieval performance. Indeed, Fig A.2 shows that excluding the NDCEM score (i.e., setting $\lambda_2=0$) leads to definitely worse EM results compared to when $\lambda_2>0$. Intuitively, a NDCEM component $\lambda_2>0$ ensures that the returned documents, in addition to containing the gold answer (thus reaching high NDCG and Top-5), are also relevant for the query, therefore preventing semantic drifts.
> To summarize, the main takeaway of Fig A.2 is precisely the fact that all the three score components are beneficial, as can be seen by the sweet spot around $\lambda_1=.2$ and $\lambda_2=.6$; using only one of them makes the agent more vulnerable to the inaccuracies of the machine reader, or traps it into non-improving states.
>
> We fixed the typo and phrasing of that paragraph.
>
> We added a section in the Appendix which explains the procedure as pseudocode for clarity.

---

### Official Review · Reviewer_NpVB · 2021-11-03

**Correctness:** 3
**Technical Novelty And Significance:** 3
**Empirical Novelty And Significance:** Not applicable
**Recommendation:** 6
**Confidence:** 3

**Main Review:**

Strengths

- Contribution in the important direction of training RL agents to find information via iterative query refinement.
- The propose of generating synthetic search sessions via Rocchi query expansions.
-  Achieving comparable performance in a more interpretable way with pretrained LMs and RL training.
- The paper is well written and easy to follow

Weaknesses

- The results of the paper might be hard to reproduce though the authors have provided enough details. I hope that the author can provide a complete training recipe so the results will be reproducible for future work.

**Summary Of The Paper:**

This paper investigates a vital direction: How can RL agents learn to use search engines to find information. The authors propose a method that leverages pre-trained models conditioned on machine reading results to guide the selection of query refinement from aggregated search documents. The method performs comparably to neural retrievers on OpenQA-NQ but operates in a more interpretable way.

**Summary Of The Review:**

This paper contributes to an interesting direction - training RL agents to find information via iterative query refinement. The proposed method achieves descent performance in a more interpretable way. But one concern is the model reproductivity. It would be very helpful to have the model full training recipe for reproductivity and benchmarking.

---

> ### Author Response · Authors · 2021-11-18
> **Response to Reviewer NpVB**
>
> We thank the reviewer for their feedback. It seems that the review largely agrees with us with respect to the paper contribution.
>
> There is only one weakness mentioned about reproducibility of the results. The paper results are fully reproducible as we open sourced the agents' code, the training and eval scripts and the trained checkpoints. In fact, we consider open sourcing of the MuZero code itself a significant contribution.
>
> The best performing agents are based on T5 and we start fine tuning from the public checkpoints. The only thing that is needed is the training data. We add a step-by-step procedure in the Appendix of the updated draft which should clarify all doubts.

---

### Official Review · Reviewer_NfBM · 2021-11-08

**Correctness:** 2
**Technical Novelty And Significance:** 2
**Empirical Novelty And Significance:** 2
**Recommendation:** 3
**Confidence:** 4

**Main Review:**

The paper proposes an interesting idea to formulate an enhanced query using a grammar, in which 3 types of operators are allowed. The grammar corresponds to those used in Lucene (and other search engines). So it is quite general.
The key strength of the paper is its exploration of this idea in query reformulation context. RL is used to learn to reformulate the query.
The paper has several weaknesses:
- While the grammar is commonly used in search engines, it is not as refined as one could expect. In pseudo relevance feedback, adding a good term into the query is only part of the task. Another part of to assign an appropriate weight to it. This second part is not taken into account in this paper, or only partially taken into account. It is assumed that a term is added with a few fixed weights. This may be suboptimal.
- The proposed method is very expensive to train. I would assume that it is also expensive in inference (this is not reported in the paper). The high cost is not rewarded at the end: it performance is (at best) similar to that of DPR. So the question is whether it is beneficial to use the expensive solution instead of a simpler one.
- The experiments show that the strategy G1 works the best, although the grammar allows to add other operators. It would be interesting to analyze why this happened. Especially, G4 contains all the operators, and a good model would be able to choose the best one among them. The fact that G4 is much worse than G1 indicates that the model is unable to make a good choice.
- The retrieval using BM25 is a strong limitation to the approach. Although BM25 is a good and efficient IR method for retrieving full documents, for question answering where one deals with much shorter documents or passages, neural models are often better.
- Reading the example provided in the paper, it does not seem that T5 or BERT has a good capability to predict the expansion term to add. Some of the added terms are not very meaningful (e.g. "3"). This may hint that the way new terms are chosen may not work well in practice. This could stem from T5 or BERT used, or from the general learning framework based on Rocchio's pseudo relevance feedback. More analysis is needed to understand the problem.
- In IR, it is often the case that pseudo relevance feedback beyond a few times (usually once) is not beneficial and may easily lead to query drift. The process of expansion contains many steps of pseudo relevance feedback. Would there a high risk of query drift?
- Finally, experiments on more datasets is desired.

**Summary Of The Paper:**

This paper proposes an approach based on MuZero to learn strategies to enhance the search query. It borrows the idea of pseudo relevance feedback of Rocchio, selects terms from the top-5 retrieved results, and incorporate a term using a predefined grammar.
The expanded query is used by BM25 to find documents.
The approach is tested on NQ dataset. It is shown to achieve a performance comparable to that of DPR.
The key contribution of the exploration of MuZero for query enhancement, combined with the idea of Rochio. This exploration is new.

**Summary Of The Review:**

+ The idea to built a structured query following a grammar is interesting. This follows the work of MuZero.
+ The use of pseudo relevance feedback to generate candidate terms to add in this context is also interesting.
- Unfortunately, the performance is not at the level expected.
- The complexity of the method is very high, making it difficult to use in practice.
- The presented grammar is used only partially in the best performing case. It is unclear if the grammar defined in this way is reasonable.

---

> ### Author Response · Authors · 2021-11-18
> **Response to Reviewer NfBM**
>
> __Grammar__ This is the first work modeling query reformulation using search operators with ML. As such, 1) opens a green field and 2) shows its potential by matching the performance of neural retrievers using only discrete retrieval operations. More sophisticated grammars could be evaluated, and, as acknowledged, the framework is general enough to support such explorations. The framework can also be combined with neural retrievers but this is beyond the current scope.
> We explicitly model weighs with the boost operator. Moreover, when the agent boosts a term already present in the refined query, its weight gets overwritten. Hence, the weight assigned to a term is not fixed and the agent can update it throughout the search episode. We limit the number of weight buckets for the sake of avoiding complexity, but obviously this is an aspect that can be further explored. We found that doubling the number of weight buckets increases headroom marginally. Thus the current modeling weights is actually quite informative and strikes a good balance with complexity.
>
> __Expensive training__ Training the T5 agent takes about 5 days on 16 TPUs, which is in line with many other seq2seq tasks, e.g. MT. Inference is quick as we run at most 19 inferences per session. The predictions are short because we only predict the next refinement. Similarly, the MuZero agent trains on 4 TPUs in about 10 days.
> Our paper proposes a new way to look at ad-hoc retrieval problems which is centered on transparency and interpretability; that value is not represented when comparing accuracy-per-compute. Furthermore, complexity can often be brought down in many ways.
>
> __Sub-policies__ We still don’t know why the sub-policy performs better than the super-policy. We hypothesize this is due to the complexity of learning multiple different policies. The ensemble experiments show there is value in combining strategies. We are investigating this issue via RL-transformers, by explicitly representing returns associated with steps, which may provide the agent with the required information to select the appropriate action.
>
> __BM25 sub optimal__ BM25 is the right framework to start investigating retrieval models that use discrete query reformulations. But, this is just the first step. We plan to combine neural retrieval and discrete operations. The performance gap between BM25 and neural retrievers is further proof of the potential of the approach, which makes an obviously inferior framework competitive with the SOTA paradigm.
>
> __Interpretability__ It is easy to imagine that the agent is trying to boost results containing numbers, which fits the question which seeks an answer with an overt quantitative aspect (“who averaged the most points…”). Of course, the agent can only express itself by means of unigrams. At the same time, the principled formulation of the paper points to straightforward ways to push the expressivity further. E.g., one could design templates for macro-actions (e.g., for digits pattern) or allow the discovery of such macros via regular expressions.  We envision future versions to be more eloquent but for the time being we need to adapt to the language the agents are empowered with (unigrams and operators).
>
> __Drift__ This is a valid concern and it is the main reason we add the ExactMatch component to the reward. This is also a broader question requiring a deeper investigation than what can be done for rebuttal. It is fair to say that this risk is present in any approach, including basic neural retrieval ones, that train on extremely sparse human signals or on heuristic signals. Grammars offer a great deal of flexibility here. Overall, this may be more of a problem with filtering actions like +/- while boosting can reverse its effect by reweighting. As we discuss in section 5.3 this is also related to notions of control.
>
> __Evals__ Evaluating on more datasets is beyond the scope of the current presentation. We agree with the reviewer and also with respect to the previous point, it is at the top of the todo list.
>
> __Expected performance__ Quantitatively speaking, the performance is comparable to that of one of the most successful current neural retrieval proposals. Qualitatively, the same level of performance is achieved only with discrete and transparent retrieval operations. This seems an impressive feat and a milestone worth sharing with the community.
>
> The complexity of the T5 agents is trivial. The MuZero framework is indeed complex, but the “pure RL" view is important to keep in sight, we believe, because it is the one that has developed the most tools to deal with active exploration in a principled way. Our work provides one of the most convincing proofs of how RL can work for hard high-dimensional NLU problems, with the practical take-home lesson that RL needs to be appropriately knowledge-infused to succeed.
>
> The grammar implements common search operators and models a subset of the query syntax defined by the search engine.

---

### Author Response · Authors · 2021-11-18
**Response to the reviewers**

We would like to thank all reviewers for their feedback and are happy to address the points that have been raised. We would like to first stress and clarify the goals and non-goals of our paper:

Our primary aim is to advance our understanding of how search, as a process, can be improved through agent-based query reformulations. This is why we have put an emphasis on transparency and interpretability. This is also why we have taken a well-established, classical retrieval method, namely BM25, as the starting point. Finally, this is why we have strived for reproducibility and made our implementation available.

Our goal - at this point - was not to maximally push the state-of-the-art, although we have been able to match the performance of a competitive neural retrieval system (DPR) by boosting a non-competitive baseline. We believe that in itself to be significant. It has also not been our goal to primarily reduce resource needs in terms of model size and compute power as we are not suggesting to deploy the system as is. We consider the neural network components of our agents, namely BERT and T5 models, to be common ingredients, available to a large base of researchers. Certainly, there are many directions for further improvements in terms of engineering and practicability. However, we did not want to over-extend the scope of the paper.

In accordance with this focus, we believe to have contributed the following insights:

- We have shown the feasibility of the “Learning-to-Search” paradigm, achieving significant improvements over the baseline and all-in-all competitive results.

- We have shown that current text representations based on transformer architectures are able to provide sufficient language modeling capabilities to support this task.

- We have contrasted the use of contemporary reinforcement learning with the emerging idea of using generative sequence-to-sequence models for reinforcement learning (e.g. Decision Transformers, (Chen et. al, 2021)), showing overall benefits of the latter. This provides strong guidance for future work. It also confirms the validity of the seq2seq approach in a novel and different domain.

- We have shown the importance of designing suitable action spaces and have presented an innovative approach, making use of rule-based grammars and search operator syntax. This can overcome the quantization problem faced in continuous action spaces.

- We have shown how to design a policy for automatically generating goal-oriented sessions based on the idea of relevance feedback and Rocchio’s algorithm. We have demonstrated that this can overcome the complexity-breakdown of methods that are purely exploration-based.

- We show that a multiobjective criterion can help address qualitative concerns without trading off top-line performance.

The reviews acknowledge that the work is novel and interesting, that the proposed framework is quite general, and that it matches the performance of one of the most popular proposals for neural retrieval (DPR) using only BM25. These are indeed some of the key takeaways from our work.

---

### Decision · Program_Chairs · 2022-01-20

**Decision:**

Reject

**Comment:**

This paper presents a method to improve search engines; the method is designed based on the BM25 retrieval method and is evaluated on NQ open dataset. The reviewers agree that the motivation is interesting and implementation is reasonable, but the authors have only showed the impact of their approach over one retrieval method and one dataset, which is limited and does not show if the method is general enough or not.